# Dynamic Infilling Anchors for Format-Constrained Generation in Diffusion LLMs

## Abstract

Diffusion large language models (dLLMs) have recently emerged as a compelling alternative to autoregressive LLMs, offering bidirectional attention and parallel sequence generation. These properties allow dLLMs to exploit global contextual information and naturally support the integration of non-sequential constraints, making them particularly suitable for format-constrained tasks such as generating parseable JSON or reasoning–answer templates. A straightforward approach is to enforce such constraints with fixed anchors, but this often results in rigid generation spans, leading to truncated reasoning or redundant content. To overcome this limitation, we propose a training-free method, Dynamic Infilling Anchors (DIA). DIA dynamically adjusts generation length by estimating appropriate end-anchor positions before content generation, followed by iterative infilling between anchors. This flexible mechanism ensures structural correctness and semantic coherence while avoiding the inefficiencies of fixed-span methods. Experiments on reasoning-oriented benchmarks demonstrate that DIA substantially improves both format compliance and answer accuracy, achieving significant gains on GSM8K and MATH under zero-shot settings. These results highlight the promise of dLLMs for reliable, structure-aware generation and establish DIA as a practical pathway toward robust format-constrained text generation.

## 1 Introduction

In recent years, diffusion large language models (dLLMs)(Nie et al., 2025; Ye et al., 2025; Labs et al., 2025; Song et al., 2025; Deepmind, 2024) have attracted increasing attention due to their distinctive computational mechanisms and promising potential. Unlike traditional autoregressive language models (AR LLMs), which rely on left-to-right sequential decoding, dLLMs are not restricted to unidirectional dependencies during generation. Instead, they employ a bidirectional attention mechanism, enabling the model to update token representations at each step by leveraging complete contextual information simultaneously. This mechanism allows all positions in a sequence to be predicted in parallel rather than generated step by step, thereby substantially enhancing both modeling flexibility and computational efficiency. Beyond efficiency gains, this parallelism also strengthens the contextual modeling capacity of dLLMs, enabling them to capture global dependencies more comprehensively.

Within this property, we identify not only the potential to enhance contextual modeling and generation efficiency, but also the possibility of directly incorporating non-sequential constraints into the generation process. The exposure of a fully masked sequence in dLLMs allows us to impose global constraints on the target output by directly editing the masked sequence. For instance, one may preemptively replace selected mask tokens with predetermined conclusions or mandatory content, thereby guiding the model toward iterative optimization under the specified requirements. This observation motivates us to explore the application of dLLMs to the problem of format-constrained generation. The term refers to scenarios in which the model's output must strictly adhere to predefined structures and requirements. For example, producing parseable JSON representations. To evaluate this capability, we adopt a representative thinking–answering task as the testing scenario, where existing dLLMs fail to achieve satisfactory outcomes.

To address these challenges, a straightforward approach is to enforce structural constraints by inserting anchors (*e.g.* $<think>$, $</think>$, $<answer>$, $</answer>$) directly into the masked

sequence. However, while this approach appears intuitive, it also introduces new challenges. Once anchor positions are fixed in advance, the generative space between them becomes rigid, forcing the model to allocate tokens within predetermined boundaries. Such rigidity can lead to suboptimal allocation of generative space and ultimately impair output quality. In practice, when the fixed span between anchors is too short, the reasoning process is often truncated before completion. On the other hand, when the span is too long, the model tends to produce redundant or repetitive content, thereby reducing both efficiency and reliability.

To obtain an appropriate generation length between anchors, thereby ensuring format correctness while maintaining generation quality, we propose a more flexible training-free strategy termed *Dynamic Infilling Anchors (DIA)*. Our approach is inspired by previous studies on dLLMs(Li et al., 2025), which demonstrates that the model can estimate the position of the end token with only one or a few prediction steps, thereby determining a suitable generation length. We extend this capability to predict the proper positions of anchors before content generation. Specifically, our method consists of two stages: (1) generation length adjustment by estimating position of the end anchor, and (2) iterative generation between fixed anchors.

The first stage of our method involves adjusting the generation space by estimating the position of the end anchor. Following the user prompt, the model initializes a relatively short, fully masked sequence, which serves as a starting point for the task output length and is dynamically extended later. For a think-answer task, this masked sequence is evenly divided into two blocks, with the corresponding begin anchors inserted at the start of each block. We then determine the anchor positions sequentially, one block at a time. Within each block, the model performs a single prediction step on the sequence, which is prefilled with the begin anchor. If the prediction fails to produce an end anchor or yields one with insufficient confidence, it suggests that the current generation length is inadequate. Therefore, we extend the block by appending additional masked tokens to ensure adequate space for content generation and repeat the prediction step. This extension continues until the model successfully produces a valid end anchor or the block length reaches its upper limit. The design of Stage I fully leverages the model's awareness of the generation space; it guarantees sufficient allocation for each phase while minimizing redundant space and unnecessary computation.

The second stage performs iterative generation after anchors are fixed. In the previous stage, we obtained a reasonable generation length and fixed the position of the end anchor. Based on this setup, we now generate the intermediate content between the anchors. This step effectively compensates for the limitations of single-step prediction and helps the model establish clear semantic boundaries across different segments, thereby promoting coherent content generation.

We validate the effectiveness of DIA on reasoning-oriented benchmarks. Experimental results on GSM8K(Cobbe et al., 2021) (0-shot) and MATH(Hendrycks et al., 2021) (0-shot) show that our method improves format correctness from 58.83% and 29.10% to **72.63%** and **76.82%**, respectively. Moreover, by better controlling the generation space, our method also improves answer accuracy from 14.86% and 19.52% to **46.78%** and **20.08%**, respectively. These results demonstrate that DIA substantially enhances both the reliability and quality of format-constrained generation with dLLMs. In summary, our contributions are three-fold:

1. We introduce a novel dLLM-based strategy for format-constrained generation.
2. We design a dynamic adjustment mechanism that flexibly allocates generative space, mitigating the rigidity of fixed-anchor methods.
3. We will release code and resources to foster reproducibility and further research in this emerging area.

## 2 RELATED WORKS

**Diffusion Large Language Models** The evolution of diffusion paradigms in language modeling can be traced back to masked language models(Devlin et al., 2019), which randomly mask a subset of tokens in the input and predict the missing content, laying the groundwork for denoising-based generation. Building on this idea, early studies introduced continuous-space diffusion language models(Jo & Hwang, 2025), mapping text into continuous latent representations and generating sequences through diffusion and reverse denoising. However, such methods suffered from ambiguity in representation and instability in decoding discrete text. To address this limitation, discrete-space

diffusion language models(Austin et al., 2023) were proposed, directly modeling diffusion and denoising at the token level, thereby aligning the process more naturally with the discrete nature of language. Along this trajectory, BlockDiffusion(Arriola et al., 2025) incorporated block-wise modeling to mitigate the computational inefficiencies of diffusion-based text generation. For large-scale pre-training, a practical strategy for diffusion large language models (dLLMs) is to initialize them from pretrained autoregressive models(Gong et al., 2025a; Ye et al., 2025) and further align them with instructions to enhance task adaptability(Yang et al., 2025b; You et al., 2025; Song et al., 2025). To strengthen advanced capabilities, researchers have also explored reinforcement learning(Wang et al., 2025; Zhao et al., 2025; Gong et al., 2025b) as a post-training method. Meanwhile, dLLMs are being extended to multimodal scenarios through cross-modal alignment, enabling broader applications in understanding and generating modalities such as images and speech.

**Format-Constraints** Format-constrained generation is critical for deploying language models, as it directly affects the parseability and reliability of code generation, structured outputs, and reasoning templates. Existing studies often constrain the input side (prompt design(Ye et al., 2024) and example-based guidance(Min et al., 2022)), yet they are unstable under long-chain or high-complexity reasoning; output-side repair (post-processing and re-ranking(Gao et al., 2025; Zhuang et al., 2025)) improves format compliance but struggles to preserve semantic and structural consistency simultaneously. Fine-tuning or reinforcement learning on task-specific data(Song et al., 2025; Xiong et al., 2023; Cui et al., 2024; Yang et al., 2023) can enhance robustness, but the approach is costly and generalizes poorly across tasks. Constrained decoding(Mündler et al., 2025; Banerjee et al., 2025) with grammars or finite-state machines enforces strict compliance at the expense of efficiency and flexibility.

**Large Language Models** The evolution of LLMs(Yang et al., 2025a; Grattafiori et al., 2024; DeepSeek-AI et al., 2025; Anthropic, 2025; Deepmind, 2025; xAI, 2025; OpenAI, 2025) has been fundamentally driven by insights from scaling laws (Kaplan et al., 2020), which reveal power-law relationships among model size, data, and compute, thereby guiding systematic capability improvements. Building on this foundation, researchers have observed the emergent phenomenon of in-context learning (ICL)(Min et al., 2022), whereby LLMs can rapidly adapt to new tasks from demonstrations without explicit parameter updates, showcasing remarkable transfer and generalization abilities. To further enhance practical usability and alignment with human preferences, post-training techniques such as fine-tuning(Ouyang et al., 2022) and reinforcement learning(Schulman et al., 2017; Rafailov et al., 2024; Shao et al., 2024) have been extensively applied, playing a central role in task adaptation and alignment. In parallel, the rise of multimodal models has spurred advances in cross-modal alignment(Li et al., 2023; Liu et al., 2023), enabling LLMs to operate effectively across text, vision, and speech, and thereby extending their versatility. Collectively, these lines of research have driven sustained progress in LLM capability, alignment, and applicability.

## 3 METHOD

### 3.1 PRELIMINARY

**Inference of dLLMs.** In the generation stage of a diffusion language model (dLLM), the response sequence to be refined is initialized by concatenating the input prompt with a fully masked sequence of a specified length:

$$x_t = \text{Concat}(\text{prompt}, \{[MASK]\}_{0:\,max\_len-1}), \tag{1}$$

where $max\_len$ denotes the fixed response length. The generation process follows a discrete-time masked diffusion procedure, which can be formulated as a Markov chain. Thus, each prediction step depends only on the previous state, and in every iteration only the masked positions are updated in parallel:

$$P_{0|t} = \prod_{s=t}^{0} \prod_{i=0}^{n-1} P_{s|s+1}(x_s^i \mid x_{s+1}), \tag{2}$$

$$P_{s|s+1}(x_s^i \mid x_{s+1}) = \begin{cases} 1 & \text{if } x_{s+1}^i \neq [M], \text{ then } x_s^i = x_{s+1}^i, \\ 1 - \max(q(x_s^i)) & \text{if } \left(x_{s+1}^i = [M], \max(q(x_s^i)) < C\right), \\ & \text{then } x_s^i = [M], \\ \max(q(x_s^i)) & \text{if } \left(x_{s+1}^i = [M], \max(q(x_s^i)) \geq C\right) \\ & \text{or } s = 0, \text{ then } x_s^i \neq [M]. \end{cases} \quad (3)$$

Where $[M]$ denotes the $[MASK]$ token, $q(x_s^i)$ represents the output logits at position $i$ in step $s$, and $C$ is the minimum confidence threshold.

## 3.2 DYNAMIC INFILLING ANCHOR

To overcome the limited flexibility of straightforward infilling methods in diffusion language models, we propose DIA, a training-free, two-stage approach. DIA selects an appropriate end-anchor position through a single-step prediction, thereby ensuring both format correctness and generation quality. The overview of our method is illustrated in Figure 1.

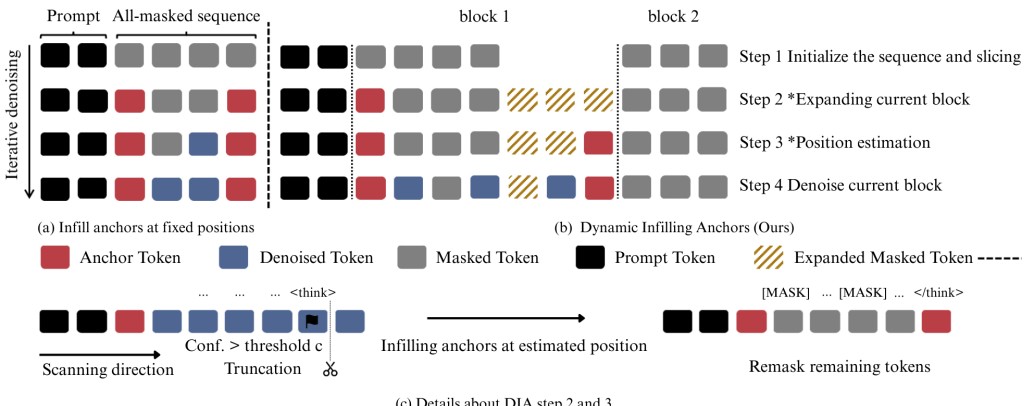

Figure 1: Dynamic Infilling Anchors (DIA). (a) Fixed-position infilling baseline. (b) Overview about our methods: DIA. (c) Details of expansion and anchor infilling steps with truncation and remasking.

### 3.2.1 GENERATION LENGTH ADJUSTMENT BY ESTIMATING POSITION OF THE END ANCHOR

DLLMs implicitly acquire a prior distribution over response termination positions from large-scale training corpora(Li et al., 2025). Specifically, for different input queries, the confidence of predicting the `eos` token at various positions within the answer sequence is not uniform, but instead exhibits a trend correlated with the appropriate response length. Building on this insight, we extend this capability to format-constrained tasks. For a typical reasoning-answer task, when the model receives the start anchor of a reasoning or answering section, it should be able to anticipate at what sequence length a corresponding "end-of-reasoning" or "end-of-answering" anchor is likely to occur. Intuitively, if the allocated generation space is sufficient to accommodate the reasoning or answering process, the one-step prediction will contain an end anchor (or partial end anchor) with high confidence exceeding a given threshold. Conversely, if the generation space is insufficient, the corresponding anchor will either fail to appear or appear only with substantially reduced confidence.

Building on this assumption, we design the generation-space estimation procedure of DIA. Given an input sequence $X$, which consists of the user query $Q$ and a fully masked sequence $X_L$ of a specified length $L$, DIA divides the sequence into two blocks ($\mathcal{C} = \{C_1, C_2\}$) of equal size (in terms of masked tokens), corresponding to the reasoning and answering stages. For each block, DIA first pre-fills the start anchor at the beginning of the decodable region. After inserting the start anchor, the block undergoes a one-step prediction. The prediction results and their associated confidence scores are used to determine whether the allocated generation length is appropriate. Since the model is

unlikely to produce a complete anchor token sequence in a single prediction, partial anchors are also incorporated into the decision mechanism. If the prediction either fails to produce an end anchor (or a partial end anchor) or yields an end anchor with confidence below the threshold $c$, the length of current block is expanded by a fixed length $\Delta$, and the 'predict–decide' cycle is repeated until the generation space is sufficient to support the model in completing the reasoning or answering process. When multiple positions in the sequence satisfy the confidence threshold simultaneously, we retain the position closest to the left boundary to prevent the generation of duplicate end anchors within the sequence. To avert unbounded expansion, a maximum block length $M$ is imposed. We truncate the redundant tokens following the selected end-anchor position and subsequently complete the partial end anchor to form a full one.

### 3.2.2 ITERATIVE DENOISING WITH INFILLING

In Stage I, we establish the block boundaries by determining the positions of the anchors. Based on these fixed semantic boundaries, the model then iteratively generates the intermediate content within the block. The fixed anchors serve as guidance, ensuring clear separation between segments and thereby promoting coherent content generation.

We process the blocks sequentially through two stages. Specifically, once the length of the thinking block is determined, its content is generated; the additional information obtained from this reasoning step is then used to determine the length of the answering block, which is subsequently generated in an iterative manner. This design maximizes the benefit of the reasoning process by leveraging the information gained in the first stage to enhance the quality of the final answer. Further implementation details are provided in Algorithm 1.

---

**Algorithm 1** DIA

---

**Require:** Input sequence $X = \{Q, X_L\}$, begin-anchor set $\mathcal{B} = \{b_1, \ldots, b_{|\mathcal{B}|}\}$, end-anchor set $\mathcal{E} = \{e_1, \ldots, e_{|\mathcal{E}|}\}$, confidence threshold $c$, expand size $\Delta$, max length $M$
**Ensure:** Completed sequence $X = \{Q, C_1, \ldots, C_{|\mathcal{B}|}\}$
  1: Divide $X_L$ into $|\mathcal{B}|$ blocks $\mathcal{C} = \{C_1, \ldots, C_{|\mathcal{B}|}\}$, each of max length $M$
  2: **for** $i \leftarrow 1$ to $|\mathcal{B}|$ **do**
  3:      Insert begin anchor $b_i$ at the head of block $C_i$
  4: **end for**
                     *Stage 1: Generation length adjustment by estimating position of the end anchor*
  5: **for** each block $C_i$ **do**
  6:      **while** True **do**
  7:          $Y \leftarrow \text{Infer}(Q, C_1 \ldots C_i)$                   *perform one diffusion-based inference*
  8:          Scan tokens of $C_i$ from head
  9:          **if** a subsequence $y \subseteq Y$ matches some part of $e_i \in \mathcal{E}$ with $\text{Conf}(y) > c$ **then**
10:             Truncate $C_i$ at this position
11:             **break**
12:          **else if** no partial match found and $|C_i| + \Delta \leq M$ **then**
13:             Expand $C_i$ by $\Delta$ tokens                  *controlled by expand size*
14:          **else if** $|C_i| + \Delta > M$ **then**
15:             Stop expansion for $C_i$
16:             **break**
17:          **end if**
18:      **end while**
19:      Infill selected $e_i \in \mathcal{E}$ at the tail of $C_i$
                            *Stage 2: Iterative Denoising with Infilling*
20:      Generate all remaining masked positions in $C_i$ using $\text{Infer}(Q, C_1 \ldots C_i)$
21: **end for**
22: **return** $X = \{Q, C_1, \ldots, C_{|\mathcal{B}|}\}$

---

## 4 EXPERIMENTS

### 4.1 BENCHMARKS

To systematically evaluate the effectiveness of our method, we adopt two reasoning-sensitive mathematical benchmarks: GSM8K 0-shot and MATH 0-shot. **GSM8K**(Cobbe et al., 2021) is a widely used dataset of grade-school math word problems, covering basic arithmetic and commonsense reasoning tasks, and thus serves as a reliable measure of a model's performance in everyday numerical reasoning scenarios. In contrast, **MATH**(Hendrycks et al., 2021) is a more challenging benchmark that spans competition-level problems from elementary to advanced mathematics, encompassing diverse problem types and difficulty levels, thereby providing a rigorous assessment of a model's capabilities in complex reasoning and knowledge generalization.

### 4.2 BASELINES

We select Dream-7B-Base-v0 and Dream-7B-Instruct-v0 as our baseline models. The Dream-7B series is initialized from the Qwen model family and has achieved superior performance compared to other open-source diffusion models on multiple benchmark tasks. To ensure fairness, all experiments are conducted with corresponding modifications to the official codebase, without applying any additional acceleration or optimization techniques.

### 4.3 IMPLEMENTATION DETAILS

Our method is implemented within the PyTorch framework. For a fair comparison, all models are evaluated under the same GPU configuration when tested on identical tasks. Additional implementation details are provided in Appendix B.

### 4.4 MAIN RESULTS

We conduct a comprehensive evaluation on the two benchmarks. Table 1 reports the comparison between our method and the baselines. Specifically, Dream-7B-Base-v0 and Dream-7B-Instruct-v0 generate responses by relying solely on additional format-constrained prompts. In contrast, the infilling approach inserts the corresponding anchors at designated positions within the response sequence of Dream-7B-Base-v0, thereby guiding the model to produce answers.

We introduce two metrics, Format Score $S_{format}$ and Accuracy $Acc.$,for evaluation. Accuracy measures whether the generated response is correct, while Format Score assesses whether the response adheres to the predefined format requirements.

Table 1: Comparison of Methods on Format Adherence and Benchmark Performance. DIA achieves the highest format scores across both GSM8K and MATH, substantially outperforming baseline and infilling approaches. These results highlight the robustness and effectiveness of DIA in enforcing strict structural constraints while maintaining competitive answer accuracy.

| | 0-shot GSM8K | | 0-shot MATH-500 | |
|---|---|---|---|---|
| | $S_{format}$ | $Acc.$ | $S_{format}$ | $Acc.$ |
| Dream-7B-Base (Ye et al., 2025) | 0 | 68.99 | 0 | 25.14 |
| Dream-7B-Instruct (Ye et al., 2025) | 0 | 15.01 | 0 | 25.28 |
| Infilling Baseline | 58.83 | 14.86 | 29.10 | 19.52 |
| **Dynamic Infilling Anchor (Ours)** | 72.63 | 46.78 | 76.82 | 20.08 |

Compared to the performance degradation introduced by the infilling approach, DIA achieves superior results in both format adherence and answer quality. On GSM8K, DIA not only raises the format score from 58.83% to 72.63% but also substantially improves accuracy from 14.86% to 46.78%, highlighting its ability to simultaneously enforce structural fidelity and enhance reasoning correctness. On the more challenging MATH benchmark, DIA boosts the format score from 29.10% to 76.82%, demonstrating remarkable robustness in preserving structural anchors even under complex

problem settings, while maintaining comparable answer accuracy to baseline methods. The results clearly demonstrate that DIA addresses the shortcomings of baseline models and methods under format-constrained tasks, ensuring accurate preservation of the required format. Moreover, unlike the infilling approach, DIA's flexible design of generation length allows each stage to maintain high answer quality, thereby achieving a better balance between performance and format correctness across diverse benchmarks.

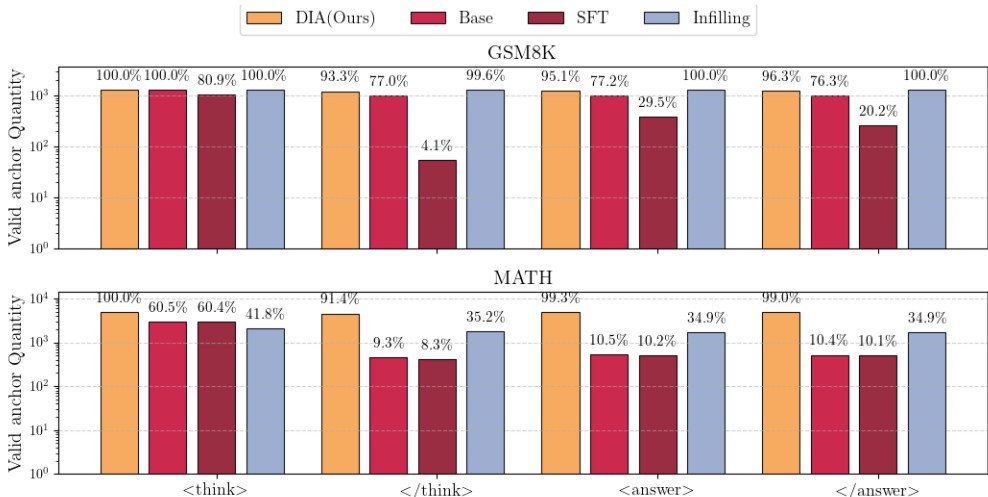

Figure 2: DIA delivers reliable anchor preservation and stable performance across different benchmarks. Even as task complexity increases on the more challenging MATH, DIA consistently maintains high anchor retention, underscoring its robustness under stricter reasoning and formatting requirements.

Figure 2 presents a detailed comparison of anchor retention ratios across different methods. Overall, DIA demonstrates outstanding stability on both GSM8K and MATH, consistently achieving nearly 100% retention across all four anchors, including both begin anchor ($<think>$ and $<answer>$) and end anchor ($</think>$ and $</answer>$). This robust performance shows that the proposed two-stage generation strategy not only preserves anchors under varying conditions but also enforces strict compliance with the predefined format throughout the entire sequence. Such stability is particularly important in reasoning-oriented tasks, where structural deviations can lead to incomplete, unparseable, or misleading outputs.

In contrast, the Base and SFT models suffer from significant structural degradation. For example, on GSM8K, their retention rates for $</think>$ collapse to only 4.4% and 29.5%, respectively, and on MATH, the rates for $</think>$ and $</answer>$ drop to single digits. These results reveal a consistent failure of conventional methods to maintain boundary integrity, especially in longer or more complex reasoning chains, where models tend to lose track of global structure and generate unbalanced outputs. Such issues undermine the reliability of the generated content and illustrate why relying solely on prompt-based constraints or fine-tuning strategies is insufficient for strict format adherence.

Although the Infilling baseline achieves higher anchor retention than Base and SFT—nearly matching DIA on GSM8K for $<think>$ and $<answer>$, its performance on begin anchors remains unstable. Crucially, this preservation does not translate into gains in overall format correctness or answer accuracy. For instance, while Infilling retains anchors on GSM8K, its downstream results remain far below DIA in both structural and semantic evaluations. This mismatch highlights that simply inserting anchors is not enough; without a dynamic mechanism for allocating and regulating generation space, models either over-generate redundant tokens or fail to stop at the correct boundaries.

Taken together, these results provide a fine-grained validation of Table 1. They show that DIA not only outperforms existing approaches in aggregate metrics but also secures overwhelming superiority in preserving critical anchors across diverse datasets. By ensuring that every anchor is faithfully

retained, DIA substantially enhances the reliability of format-constrained generation, laying a foundation for robust application of dLLMs in reasoning, structured reporting, and other scenarios where strict adherence to format is essential.

## 4.5 ANALYSIS EXPERIMENTS

### 4.5.1 BEHAVIOR OUTSIDE ANCHOR CONTEXTS

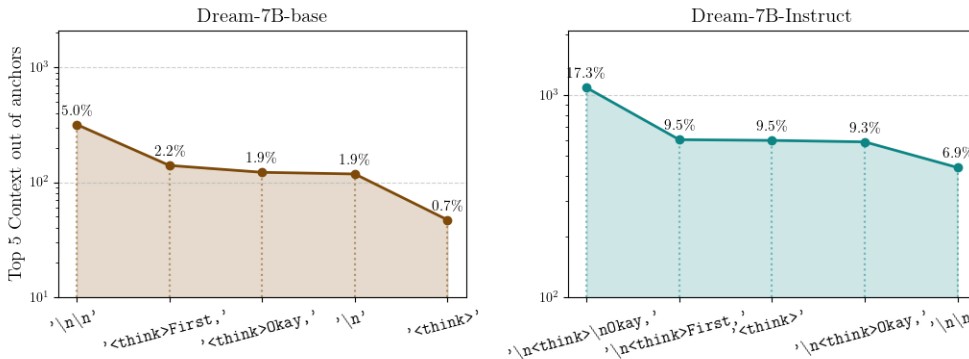

Figure 3: Top-5 statistics of out-of-anchor content generated by baseline models across different benchmarks. The baseline models fail to establish effective semantic boundaries aligned with anchor positions, leading to unconstrained content generation.

To more comprehensively evaluate model performance under format-constrained tasks, we conducted a statistical analysis of the responses generated by Dream-7B-Base-v0 and Dream-7B-Instruct-v0 on two benchmarks. Our analysis focuses on the content appearing beyond the $</answer>$ anchor boundary, as this indicates whether the models can effectively leverage the semantic boundaries established by anchors to properly constrain their generation. Specifically, we examined all responses across the two benchmarks and extracted the Top-5 most frequent continuations occurring after the $</answer>$ boundary for both models, with the results shown in Figure 3. This analysis provides a fine-grained perspective on boundary robustness, complementing aggregate metrics by revealing how models behave when the intended termination point has already been reached.

As shown in Figure 3, Dream-7B-Base-v0 produces dispersed and low-frequency redundancy beyond the $</answer>$ anchor, with all Top-5 patterns below 6%, whereas Dream-7B-Instruct-v0 exhibits more concentrated redundancy, with Top-5 patterns reaching up to 17.3% and dominated by repeated $<think>$ tokens. The contrast highlights that the Base model tends toward uncontrolled drifting, while the Instruct variant systematically re-enters the reasoning phase, reflecting a structural weakness in anchor boundary enforcement. Overall, the Base model lacks effective boundary control, while the Instruct model suffers from patterned continuations, and both fail to reliably terminate at the anchor—underscoring the necessity of DIA in eliminating out-of-anchor redundancy and ensuring format adherence. Importantly, such failures not only compromise readability but also propagate errors to downstream applications that rely on strictly bounded outputs.

### 4.5.2 EXPAND TIMES

To establish a reasonable upper bound for the maximum block length, we analyzed the number of extensions in the reasoning part of all format-correct responses. The details are presented in Figure 4. The results show that the chosen maximum length threshold effectively ensures the allocation of appropriate generation space. Specifically, the observed extension counts fall within the range of (30, 85), which is substantially smaller than the number of extensions permitted by the maximum threshold. In other words, although a large upper bound is allowed, the vast majority of responses naturally converge to a much smaller range of expansions, confirming that the setting of the maximum block length is both sufficient and not overly restrictive.

Moreover, the proportion of format-correct responses within this range consistently exceeds 90%, further validating both the effectiveness of our method and the appropriateness of the current threshold setting. Importantly, this trend is observed across both GSM8K and MATH, where over 94% of samples fall into the effective expansion range, indicating that DIA adapts reliably to tasks of different scales and difficulties. Such stability suggests that the block-length constraint not only prevents degenerate over-expansion but also preserves high-quality structural adherence across benchmarks.

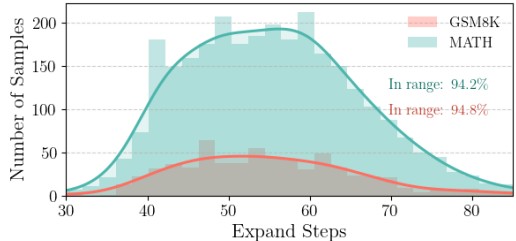

Figure 4: Statistics of effectively expanded samples. The maximum length threshold ensures that the vast majority of cases receive an appropriate number of expansions, thereby safeguarding answer quality.

## 5 DISCUSSIONS

While Dynamic Infilling Anchors (DIA) have
shown strong effectiveness in improving format-constrained generation, several limitations remain. First, the current method relies on manually specified anchors, assuming that task boundaries and semantic roles (*e.g.*, reasoning vs. answering) are fixed. In more complex tasks such as open-domain dialogue or multi-stage reasoning, anchors may not follow stable positions, and their semantics may shift with context. Extending DIA to automatically infer or adapt anchor definitions remains an open challenge. Second, DIA introduces inference overhead from iterative space adjustments. Although modest in our experiments, scaling to longer outputs or interactive systems may require more efficient mechanisms for anchor prediction and length control. Finally, our evaluation is limited to reasoning datasets like GSM8K and MATH; whether DIA generalizes to long-form writing, program synthesis, or multimodal tasks is still uncertain.

Despite these limitations, the anchor-based perspective also suggests new opportunities. Anchors need not be restricted to reasoning and answering boundaries; they could represent higher-level structures such as proof steps, code blocks, or multimodal transitions. In tasks where anchor semantics evolve dynamically, DIA could be extended with adaptive or hierarchical anchor systems that refine themselves during generation. Combining DIA with lightweight training or alignment methods may also help models acquire richer anchor semantics, reducing manual specification while improving robustness across domains. These possibilities suggest that anchors are not only a control mechanism for current benchmarks but also a lens for rethinking how diffusion language models manage structure and meaning in more complex generative scenarios.

## 6 CONCLUSION

In this work, we introduced Dynamic Infilling Anchors (DIA), a training-free method for enhancing format-constrained generation in diffusion language models (dLLMs). By adopting a two-stage decoding strategy—length expansion guided by end-anchor prediction, followed by explicit anchor completion and content generation—DIA achieves a strong balance between structural fidelity and semantic quality. Experiments on reasoning benchmarks such as GSM8K and MATH show that DIA substantially improves format adherence while maintaining competitive answer accuracy, validating its effectiveness in tasks requiring both precision and reliability.

Beyond these empirical results, our study underscores the broader potential of dLLMs for structured text generation without additional training. Leveraging their intrinsic awareness of generation space, DIA demonstrates that such models can be steered to meet strict output constraints, opening opportunities in code generation, structured proofs, and machine-readable reporting. At the same time, DIA points to several avenues for future research, including automated anchor design, extensions to hierarchical or nested constraints, and integration with lightweight training or alignment techniques. By bridging structural control and semantic quality, DIA lays the groundwork for deploying dLLMs as dependable reasoning assistants and as general-purpose models in real-world applications where consistency, interpretability, and reliability are paramount.

ETHICS STATEMENT

This research does not involve human participants, personally identifiable information, or sensitive user data. All experiments are carried out on publicly available datasets, namely GSM8K and MATH, which are widely adopted benchmarks for assessing the reasoning ability of large language models. These datasets consist of synthetic or anonymized problem–solution pairs and do not contain private or proprietary information, thereby avoiding risks associated with data collection or misuse.

As with any work involving large language models, we acknowledge potential ethical concerns. A risk of misuse remains: models augmented with improved format-adherence mechanisms could be applied in contexts where reasoning outputs are consumed without adequate verification, potentially leading to the dissemination of incorrect or misleading content. We emphasize that our proposed method should only be deployed in scenarios where outputs are subject to human oversight or rigorous automated validation. While our method focuses on structural control rather than altering model parameters, it inherits the biases and limitations of the underlying pretrained models. Although Dynamic Infilling Anchors (DIA) does not amplify such issues, users of this technique should remain mindful of the broader ethical challenges associated with large-scale language models.

We stress the importance of responsible deployment. The intention of this work is to advance research on reliable, format-constrained generation and to provide the community with a training-free mechanism that improves structural fidelity without additional fine-tuning or data collection. We do not endorse its use in high-stakes domains such as medical diagnosis, legal reasoning, or automated decision-making without extensive domain-specific evaluation and safeguards. By situating our contributions within established ethical guidelines in AI research, we aim to ensure that the benefits of this work are realized while minimizing risks of harm.

REPRODUCIBILITY STATEMENT

We have taken comprehensive steps to maximize the reproducibility of our results and to facilitate independent verification by other researchers. To this end, we adhere to several principles of transparent and replicable research practice.

The appendix provides extensive details about our experimental setup, including hardware configuration (GPU model, number of devices, and memory constraints), software environment (Python and PyTorch versions, dependencies, and library compatibility), parameter choices (generation hyperparameters, block sizes, anchor thresholds, and sampling strategies), and evaluation protocols (dataset splits, metrics, and scoring procedures). These details are presented to eliminate ambiguity and enable replication.

The implementation of Dynamic Infilling Anchors (DIA) is based on the official Dream-7B codebase with minimal modifications, all of which are clearly documented. We describe both algorithmic adjustments, such as the two-stage anchor-based procedure for length expansion and truncation, and essential engineering decisions including input preprocessing and memory optimization strategies. Such documentation lowers the barrier for others to replicate our findings without requiring extensive reverse engineering.

We also commit to releasing all necessary artifacts for reproduction. This includes the source code, scripts for data preprocessing and evaluation, and configuration files for running experiments. The release will be accompanied by instructions for environment setup and guidelines for reproducing results on GSM8K and MATH. Seed values and randomization settings will be provided to ensure consistent outputs across trials.

In line with community standards, long-term accessibility is emphasized. All released materials will be archived in a permanent public repository with version control, ensuring that they remain accessible as dependencies evolve. By combining detailed documentation, transparent reporting, and open resources, we aim to make our work fully reproducible and to encourage further extensions and critical evaluation by the research community.

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

## A  Notation Summary

Table 2: Notation Summary

| Symbol | Description |
|---|---|
| $X = \{Q, X_L\}$ | Input sequence consisting of query $Q$ and all-masked sequence $X_L$ |
| $Q$ | Input query provided to the diffusion-based LLM |
| $X_L$ | Fully masked sequence to be partitioned into blocks |
| $\mathcal{B} = \{b_1, \ldots, b_{|\mathcal{B}|}\}$ | Set of begin anchors inserted at block heads |
| $\mathcal{C} = \{C_1, \ldots, C_{|\mathcal{B}|}\}$ | Set of blocks obtained from $X_L$ |
| $\mathcal{E} = \{e_1, \ldots, e_{|\mathcal{E}|}\}$ | Set of end anchors to be infilled at block tails |
| $C_i$ | The $i$-th block after begin anchor insertion and expansion |
| $|C_i|$ | Current length of block $C_i$ during expansion |
| $y$ | Predicted subsequence scanned within block $C_i$ |
| $c$ | Confidence threshold for partial end-anchor detection |
| $\Delta$ | Expand size: number of tokens added per expansion step |
| $M$ | Maximum allowed length of a block $C_i$ |
| $\mathrm{Infer}(X)$ | Diffusion-based inference function applied to sequence $X$ |
| $\mathrm{Conf}(y)$ | Confidence score assigned to subsequence $y$ |

## B  Experimental Settings

Table 3: Experimental Settings

| Category | Setting |
|---|---|
| Model | Dream-7B-Base / Dream-7B-Instruct |
| Datasets(Max new tokens) | GSM8K(256), MATH-500(512) |
| Evaluation Metrics | Format Score, Answer Score, Anchor Retention Rate |
| Confidence Threshold $c$ | 0.065,0.05 |
| Expand Size $\Delta$ | 4,4 |
| Max Block Length $M$ | 512,512 |
| Diffusion Steps | 512,512 |
| Batch Size | 1,3 |
| Total Samples | GSM8K: 1319, MATH-500: 5000 |
| Hardware | NVIDIA vGPU-32G, vGPU-48G |
| Framework | PyTorch 2.5.1, Python 3.10.16 |

## C  Case Study

### C.1  Base Model

```
"query":  "Brandon's iPhone is four times as old as Ben's iPhone.
Ben's iPhone is two times older than Suzy's iPhone.  If Suzy's
iPhone is 1 year old, how old is Brandon's iPhone?", "method":
"Base", "output":  "<think> Suzy's iPhone is 1 year old.  Ben's
iPhone is two times older than Suzy's iPhone, so Ben's iPhone is 2
years old.  Brandon's iPhone is four times as old as Ben's iPhone,
so Brandon's iPhone is 4 years old.  So the answer is 4 years
old.</think><answer> 4 </answer>\n\n", "flaw":  "Content out of
anchor"
```

```
"query":  "Well's mother sells watermelons, peppers, and oranges
at the local store.  A watermelon costs three times what each
pepper costs.  An orange costs 5 less than what a watermelon cost.
Dillon is sent to the store to buy 4 watermelons, 20 peppers, and
10 oranges.  What's the total amount of money he will spend if
each pepper costs 15$?", "method":  "Base", "output":
"<<think>First, I need to find the cost of a watermelon.  Since a
watermelon costs three times what each pepper costs, and each
pepper costs 15$, I can multiply 15 by 3 to get the cost of a
watermelon.  15 * 3 = 45.  So a watermelon costs 45$.\n\nNext, I
need to find the cost of an orange.  Since an orange costs 5 less
than what a watermelon costs, and a watermelon costs 45$, I can
subtract 5 from 45 to get the cost of an orange.  45 - 5 = 40.  So
an orange costs 40$.\n\nNow, I need to find the total amount of
money Dillon will spend.  He is buying 4 watermelons, 20 peppers,
and 10 oranges.  I can calculate the total cost by multiplying the
cost of each item by the number of items and adding them
together.\n\nFor the watermelons:  4 * 45 = 180\nFor the peppers:
20 * 15 = 300\nFor the oranges:  10 * 40 = 400", "flaw":  "Missing
anchors"
```

## C.2   INSTRUCT MODEL

```
"query":  "Peter plans to go to the movies this week.  He always
gets a ticket for $7 and popcorn for $7.  If he has 42 dollars for
the week, how many times can he go to the movies?", "method":
"Instruct", "output":  "<think><answer>3 times</answer>", "flaw":
"Missing anchors.  Incomplete response."
```

```
"query":  "Peter plans to go to the movies this week.  He always
gets a ticket for $7 and popcorn for $7.  If he has 42 dollars for
the week, how many times can he go to the movies?", "method":
"Instruct", "output":  "" <think>$21</think>"", "flaw":  "Missing
anchors.  Incomplete response."
```

## C.3   INFILLING METHOD

```
"query":  "Peter plans to go to the movies this week.  He always
gets a ticket for $7 and popcorn for $7.  If he has 42 dollars for
the week, how many times can he go to the movies?", "method":
"Infilling method", "output":  "" <think>...</think>
<answer>....</answer> <think>.</think> <answer>....</answer>
<think>....</think> <answer>....</answer> <think>....</think>
<answer>....</answer> <think>....</think> <answer>....</answer>
<think>....</think> <answer>....</answer> <think>....</think>
<answer>....</answer> <think>....</think> <answer>....</answer>
<think>....</think> <answer>....</answer>
<think>....</think><answer>....</answer> <think>....</think>
<answer>....</answer> <think>....</think> <answer>....</answer>
<think>....</think> <answer>....</answer> <think>....</think>
<answer>....</answer> <think>....</think> <answer>....</answer>
<think>....</think> <answer>....</answer>
<think>....</think><think>....</think> <answer>....</answer>
<think>....</think> <answer>....</answer>"", "flaw":  "Broken
response."
```

```
"query":  "Kyle bought last year's best-selling book for $19.50.
This is with a 25% discount from the original price.  What was the
original price of the book?", "method":  "Infilling method",
"output":  "" <think>25</think> <answer>25</answer> \nThe
question:\n"", "flaw":  "Content out of anchor.  Incomplete
response."
```

## C.4 DIA

"query": "Charlie wants to sell beeswax candles. For every pound
of beeswax, he can make 10 tapered candles. One pound of beeswax
and the wicks cost $10.00 in supplies. If he sells each candle
for $2.00 each, what is his net profit if he makes and sells 20
candles?", "method": "DIA", "output": ""<think>First, I need to
figure out how much candles Charlie can make with 20 candles.
Since he can make 10 candles from one pound of beeswax, he can
make 20 candles from 2 pounds of beeswax. The pound of beeswax
costs $10, so 2 pounds will cost $20. The total of supplies is
$20. Now, I need to calculate out how much money he will make
from selling 20 candles. Since he sells each candle for $2, he
will make 20 x $2 = $40. Now, I need to subtract the cost of
supplies from the amount he will make from selling the candles.
So, $40 - $20 = $20. Therefore, Charlie will profit a net of
$20.00 and makes and sells 20 candles0</think><answer>
20</answer>"", "flaw": "None."

"query": "John drives for 3 hours at a speed of 60 mph and then
turns around because he realizes he forgot something very
important at home. He tries to get home in 4 hours but spends the
first 2 hours in standstill traffic. He spends the next half-hour
driving at a speed of 30mph, before being able to drive the
remaining time of the 4 hours going at 80 mph. How far is he from
home at the end of those 4 hours?", "method": "DIA", "output":
""<think>First, I need to calculate the distance John traveled
away. He drove for 3 hours at 60 mph, so the distance is 60 * 3 =
180 miles. Then, he turned around and tried to get home in 4
hours. He, spent the first 2 hours in standstill traffic, so he
didn't cover any distance during that time. He, then spent the
next half-hour driving at 30 mph, so the covered distance is 30 *
0.5 = 15 miles. He, spent the remaining 2 hours driving at 80
mph, so the covered distance is 80 * 2 = 160 miles. The, the
total distance he covered while coming back is 15 + 160 = 175
miles. Since, he traveled 180 miles away from home and then
covered 175 miles back,,, he is distance from home at 180 - 175 =
5 miles. the end of those 4 hours, he, he, 55555555555555
miles5</think><answer>5 miles</answer>"", "flaw": **"Generation
length prediction not completely accurate."**

## D  DLLM PROMPT TEMPLATE

Table 4: Example of Prompt Design

| Field | Content |
|---|---|
| Instruction | You are a helpful assistant that helps the user to solve the question. |
| Output Format | You need to think first and then answer the question briefly by following the format:$< think > ... < /think >< answer > ... < /answer >$. |
| Input | Here are the questions: {QUESTION} |

## E  USAGE OF LLMS

In accordance with the ICLR guidelines, we disclose the use of large language models (LLMs) in
the preparation of this paper. LLMs were employed exclusively as a writing assistance tool for
language polishing, grammar refinement, and improving readability. They were not involved in
research ideation, experimental design, data analysis. All technical ideas, theoretical developments,

proofs, and experimental results presented in this paper are entirely the work of the authors. The authors take full responsibility for the accuracy and integrity of the final submission.

