# OpenReview forum: "Dynamic Infilling Anchors for Format-Constrained Generation in Diffusion LLMs"
_ICLR.cc/2026/Conference — ICLR 2026 Conference Withdrawn Submission_

### Official Review · Reviewer_kFiS · 2025-10-31

**Soundness:** 2
**Presentation:** 2
**Contribution:** 2
**Rating:** 2
**Confidence:** 2

**Summary:**

The paper examines diffusion LLMs (dLLMs), and specifically their ability to produce output that strictly meets a required format, such as JSON.  It proposes an approach to inserting format anchors that meet these formatting requirements, Dynamic Infilling Anchors (DIA), with two stages; the first stage involves estimating positions of anchors, and the second stage fills content iteratively based on anchor location.  Evaluation involves comparing DIA with two dLLM baselines, Dream-7B-Base-v0 and Dream-7B-Instruct-v0, as well as an infilling baseline, with an analysis of both formatting correctness and accuracy on two well-known datasets, GSM8K and MATH.

**Strengths:**

* dLLMs look quite promising but there is still much to be explored about them, in particular in terms of addressing existing shortcomings, and this paper contributes to that.

**Weaknesses:**

The major issues I had were with respect to the evaluation:

* S_{format} isn’t defined precisely.  It’s difficult to infer exactly what its definition should be, and why the values for the baselines on both datasets in Table 1 are 0.  How do those values relate to the (non-zero) numbers in Fig 2 and discussion in Sec 4.4?

* Similarly, accuracy isn’t defined.  I assume it’s accuracy over all data items, not some subset, e.g. the set of all correctly formatted responses.

* Relatedly, the operation of the baselines isn’t explained.  What were the prompts for the two Dream dLLMs?  Is it the prompts in App D? This is not linked from Sec 4.2, and the caption just says that they are example prompts.  (Also, please give citation in Sec 4.2 for the baselines – presumably this is just the Ye et al paper.)  The exact prompt and how it specifies would influence judgments about the extent to which this is a fair comparison and what the comparison means -- I would have expected some discussion of the work in autoregressive models on generating structured outputs, e.g. [GC+25,LL+24], and related implications for what's necessary in terms of prompts for good performance on structured output.  Also, what is the infilling baseline?  As far as I can see, it’s just defined by the diagram in Fig 1(a).

* While I recognize that the primary purpose of the work is to improve dLLMs and assess the improvement in that context, I would have expected to see some autoregressive baselines here also to calibrate overall performance.  The DeepMind model cited in the first paragraph of Sec 1, for example, has such a comparison, including over reasoning / math tasks like the present paper.

* The DIA method involves iteration of stages, and it’s unclear whether this introduces a significant extra cost or not.  What effect does this have on performance time-wise?


Minor points:

* What is “SFT” in Sec 4.4 and Fig 2?  Is it Dream-7B-Instruct-v0?

* I think the value 4.4% in Sec 4.4 should be 4.1%, but I may be misreading the relevant part of Fig 2.


[GC+25] Geng et al (2025).  JSONSchemaBench: A Rigorous Benchmark of Structured Outputs for Language Models. arXiv.

[LL+24] Liu et al (2024). Are LLMs good at structured outputs? A benchmark for evaluating structured output capabilities in LLMs.  Information Processing & Management.

**Questions:**

Please see above.

---

### Official Review · Reviewer_S2H2 · 2025-10-31

**Soundness:** 3
**Presentation:** 3
**Contribution:** 2
**Rating:** 4
**Confidence:** 4

**Summary:**

The paper introduces a method for dynamically determining generation length using infilling anchors during decoding. Instead of relying on a fixed max length or external stopping heuristic, the model learns adaptive anchor tokens that signal completion. The approach is tested on reasoning benchmarks such as GSM8K, MATH, and StrategyQA, showing competitive results with more consistent output formatting.

**Strengths:**

- Simple and intuitive method for adaptive generation length control.
- Well-motivated idea that avoids complex heuristics or external control modules.
- Demonstrates improvements in formatting consistency for reasoning tasks.
- Clear visual explanation of anchor-based decoding.

**Weaknesses:**

-Evaluation is limited to reasoning datasets, which are not heavily formatting-dependent. Structured or syntax-sensitive tasks — such as JSON generation, text-to-SQL (Spider), or code synthesis (HumanEval, MBPP) — would provide a stronger testbed. See also “Generating Structured Outputs from Language Models: Benchmark and Studies” (Arxiv:2501.10868) for relevant comparisons.
- No competitive constrained-decoding baselines for comparison.
- Accuracy drops on GSM8K and MATH (Table 1) raise questions about whether formatting gains trade off against reasoning quality.
- Missing deeper efficiency analysis — e.g., compute cost, FLOPs, or latency improvements are not discussed.
- Excessive background on discrete diffusion models, which feels tangential.
- NIT : Figure 1(a) and 1(b) are placed too close together, appearing as a single approach — better spacing or captions could improve readability.

**Questions:**

- Can the method generalize to structured or syntax-sensitive domains such as JSON or code generation? (See Weaknesses)
- How does anchor-based decoding affect latency or decoding speed?
- Could the authors add comparisons with SOTA constrained decoding methods for DLMs?

---

### Official Review · Reviewer_fGWZ · 2025-11-01

**Soundness:** 2
**Presentation:** 3
**Contribution:** 2
**Rating:** 4
**Confidence:** 3

**Summary:**

This paper proposes Dynamic Infilling Anchors (DIA), a training-free method to improve format-constrained generation in diffusion large language models (dLLMs). The approach addresses the limitation that dLLMs struggle to generate outputs adhering to predefined structures (e.g., reasoning-answer templates with <think>...</think><answer>...</answer> format). The key innovation is a two-stage method: (1) dynamically estimating appropriate end-anchor positions through single-step prediction before content generation, and (2) iteratively infilling content between fixed anchors. The method is evaluated on GSM8K and MATH benchmarks in zero-shot settings, showing significant improvements in both format compliance (from 58.83% to 72.63% on GSM8K, from 29.10% to 76.82% on MATH) and answer accuracy (from 14.86% to 46.78% on GSM8K).

**Strengths:**

1. Format-constrained generation is crucial for deploying dLLMs in real applications. The paper identifies a genuine limitation where existing dLLMs fail to reliably produce structured outputs.
2. The training-free feature and implementation on top of existing dLLM codebases makes DIA easy to adopt. The promise to release code is good for repro and applications..
3. The improvements in both format compliance and answer accuracy on GSM8K are impressive, particularly recovering from the performance degradation of naive infilling.
4. Section 4.5 provides valuable insights through analysis of out-of-anchor content and expansion statistics, helping understand when and why the method works.
5. The appendix provides extensive details including case studies, prompt templates, and experimental settings that help reproducibility.

**Weaknesses:**

1. The paper only compares against a fixed-position infilling baseline and base models. Critical comparisons are missing, include recent dLLM constrained generation work like Constrained Discrete Diffusion, constrained decoding frameworks like XGrammar, and Post-processing methods that fix format violations. These comparisons are essential to establish whether the problem requires a dLLM-specific solution.

2. The experiments scope is narrow and in small scale. This paper only includes two benchmarks (GSM8K, MATH), both mathematical reasoning. Those experiments are all using a single model family (Dream-7B, only 7B scale) and with only think-answer format despite broader claims. There is no evaluation on more diverse structured tasks like code generation, JSON/structured output, multi-stage reasoning, longer documents as well as no comparison across different dLLM architectures

3. For a method adding iterative expansion steps, the paper provides no walltime measurements, memory usage analysis, or comparison of inference cost vs. baselines. The abstract claims DIA provides "reliable, structure-aware generation" but doesn't quantify the overhead cost.

**Questions:**

1. Can you provide more details on the fixed-position infilling baseline? Why does it cause such dramatic performance degradation (68.99% -> 14.86%)? This seems unusually high that we need better understanding of why it fails.

2. How were the confidence thresholds (c=0.065 for GSM8K, c=0.05 for MATH) selected? Can you provide ablation studies showing sensitivity to this parameter? How should those parameters be set this for new tasks?

3, Computational overhead: What is the actual inference latency and memory usage of DIA compared to standard generation? What is the wall-clock time comparison?

4. Have you evaluated DIA on other dLLM architectures beyond Dream-7B? Can you show results on non-mathematical tasks (e.g., JSON generation, code with specific formats)?

5. Comparison with important baselines mentioned in the weakness.

---

### Note · Authors · 2026-01-05

I have read and agree with the venue's withdrawal policy on behalf of myself and my co-authors.